# Eccentric Compression Properties of FRP–Concrete–Steel Double-Skin Square Tubular Columns

**DOI:** 10.3390/polym15122642

**Published:** 2023-06-10

**Authors:** Dai Wang, Jiansong Yuan, Jiahua Jing, Chengrui Fu, Yuhang Wang, Jiaru Xiong

**Affiliations:** 1Assessment Technique Research Center of Civil Engineering, Zhengzhou University of Technology, Zhengzhou 450044, China; 20071060@zzut.edu.cn (D.W.); 15331993285@163.com (C.F.); 2College of Civil Engineering, Henan University of Engineering, Zhengzhou 451191, China; 13733686535@163.com (Y.W.); 18238535670@163.com (J.X.); 3School of Civil Engineering and Architecture, Anyang Normal University, Anyang 455000, China; jingjiahua@aynu.edu.cn

**Keywords:** FRP–concrete–steel, tubular columns, eccentric compression properties, experimental study

## Abstract

FRP (fiber-reinforced polymer)–concrete–steel double-skin square tubular (FCSST) columns are composed of an outside FRP tube, an inside steel tube and the concrete filled between them. Under the continuous constraint of the outside and inside tube, the strain, strength and ductility of concrete are improved significantly compared with those of traditionally reinforced concrete without lateral restraint. Additionally, the outside and inside tube not only function as the permanent formwork in casting but improve the bending and shear resistance of composite columns. Meanwhile, the hollow core also reduces the weight of the structure. Through the compressive testing of 19 FCSST columns subjected to eccentric load, this study focuses on the influence of eccentricity and layers of axial FRP cloth (away from the loading point) on the evolution of axial strain along the cross-section, axial bearing capacity, axial load–lateral deflection curve and other eccentric properties. The results can provide basis and reference for the design and construction of FCSST columns and are of great theoretical significance and practical value for the application of composite columns in the engineering of structures in a corrosive environment and other harsh conditions.

## 1. Introduction

Hollow-core concrete columns are widely used in high bridge piers, especially in seismic zones. The application of a hollow core can reduce the weight of the column, which thereby decreases the proportion of the self-weight of structures in designed force. Compared with traditional concrete columns reinforced by steel bars or tubes, concrete-filled double-skin steel tube columns present superior performances. Numerous studies [1,2,3] have revealed that steel tubes not only function as a permanent formwork during casting but increase the compressive and shear resistance of columns. Meanwhile, the constant constraint of a steel tube can improve the ductility and strength of the concrete. The core concrete can also delay the local buckling of the steel tube. 

The FRP (fiber-reinforced polymer)–concrete–steel tube column, as a new core column, combines the advantages of its three individual materials: FRP, concrete and a steel tube. In terms of columns subjected to compressive, bending and horizontal reciprocating loads, the FRP used for the columns of structures has better specific strength and superior corrosion resistance than a steel tube [4,5,6,7,8].

The structural behaviors of an FRP–concrete–steel double-skin tubular column subjected to axial compression [9,10,11,12] and bending load [13,14] have been investigated. Liao et al. [10] examined the axial compressive behavior of an FRP-confined seawater sea-sand concrete-filled stainless steel tubular column. Zhang et al. [13] studied the flexural behavior of rectangular hybrid FRP–concrete–steel hollow beams with RAC (recycled aggregate concrete). These test results showed that the strength and ductility of core concrete are greatly improved with the confinement of both the inner and external tubes. Shi et al. [15] conducted experimental and numerical investigations of the seismic performance of a novel self-centering hollow-core fiber-reinforced polymer–concrete–steel bridge column. Similarly, Zhang et al. [16] studied the cyclic behavior of a rectangular FRP–concrete–steel double-skin tubular column under a combined loading condition of axial compression and cyclic lateral loading. Xie et al. [17,18] examined large-scale hybrid FRP–concrete–steel double-skin tubular columns of varying slenderness ratios under concentric or eccentric compression. Zeng et al. [19,20] studied the behavior of FRP–concrete–steel double-skin tubular columns with high-strength concrete and a rib-stiffened high-strength steel tube under axial compression and seismic loadings. These findings further verified the excellent performance of FRP–concrete–steel double-skin tubular columns under different loading schemes. 

Among these columns, the circular cross-section is the most popular. For example, Abdelkarim et al. [21] explored the behavior of composite circular columns under both axial compression and cyclic horizontal force; Zhang et al. [22] carried out a systematic experimental study on hybrid FRP–concrete–steel double-skin tubular columns under two types of loading schemes: a single unloading/reloading cycle and repeated unloading/reloading cycles. More recently, Pavithra et al. [23] investigated the axial compression behavior of hybrid composite FRP–concrete–steel double-skin tubular columns with various fiber orientations, and the main variables included the orientation of the fiber angle, thickness of the FRP tube, concrete strength and void ratio of the steel tube. 

As mentioned above, the composite of FRP, concrete and steel adopted in columns can take full advantage of those positive performances in columns. Accordingly, this composite of FRP–concrete–steel in columns has great mechanical behavior, especially for columns under eccentric compression. Therefore, this column still has huge potential for application, i.e., for piers in a corrosive environment, support in industrial and civil buildings, etc. [15,21,24,25]. Additionally, its good seismic resistance could guarantee its application in building structures with seismic requirements [26,27,28,29,30]. 

However, considering the reliable beam–column connection, square columns [16,31] are being given more attention than circular columns in some structures. Moreover, the existing research mainly focused on the axial compression [4,6,7,8,9,10,11,12,19,22,32,33] and seismic performance [15,16,20,21] of columns. The findings regarding the compression and bending performance of hollow-core square columns are still limited, which has impeded the application of this new structure in engineering. Therefore, this study explores the performance of FRP–concrete–steel double-skin tubular columns subjected to eccentric loading. The number of axial FRP cloth layers and the eccentricity are considered. Especially for the purpose of low cost, the FRP cloth is only used on the tension side in this test, and its layers increase with the increasing eccentricity. Moreover, a larger range of relative eccentricity is designed in this study, which is different from most studies on relatively small relative eccentricity [5,34,35]. The test process and results of the study aim to provide reference for the design and engineering application of FRP–concrete–steel double-skin tubular columns. 

## 2. Experiment

### 2.1. Materials and Specimen Design

Since glass-fiber-reinforced polymer (GFRP) has the advantages of low cost, easy fabrication and high tensile performance, it was used for the FRP cloth in this study. The outer two-way FPR cloth (Nanjing Hitech Composites Co., Ltd., Nanjing, China) and inner circular steel tube (Sanxi Linyi Sanyuan Tube Industry Co., Ltd., Linyi, China) were used as the reinforcement materials of a square tubular concrete column named FCSST (FRP–concrete–steel double-skin square tubular) column. The thickness of the FPR cloth was 0.17 mm. The tensile strength and elastic modulus of FRP, tested in accordance with GB/T 1446-2005 [36] and GB/T 3354-2014 [37], were 2650 MPa and 160 GPa, respectively. The yield strength and elastic modulus of the steel tube, tested in accordance with GB/T228.1-2010 [38] and GB/T 2975-2018 [39], were 430 MPa and 206.2 GPa, respectively. The cubic compressive strength, tested (Testing machine: Xinsansi (Shanghai) Enterprise Development Co., Ltd., Shanghai, China) in accordance with GB/T 50081-2019 [40], is listed in Table 1 for each specimen.

A total of 19 columns of 5 series with varying eccentricities and layers of FRP cloth on the tension side (see Table 1) were designed to explore their eccentric compression performances.

For all columns, as schematically shown in Figure 1, the height was 500 mm; the cross-section size of the square concrete was 150 mm in side length and 20 mm in chamfering radius; the cross-section size of the steel tube was 76 mm in diameter and 4 mm in thickness. 

### 2.2. Specimen Preparation

Strain gauge (Hebei Xingtai Jinli Sensing Element Factory, Xingtai, China) attachment to steel tube. Before pouring the concrete, strain gauges 10 mm in length were attached on the outer surface of steel tube, especially for those columns whose eccentricity was not 0 mm. The two opposite axial strain gauges were attached on the mid-span surface of the steel tube to make sure that one could measure the axial compressive strain, and the other one could measure the axial tensile strain (see Figure 2). 

Preparing the formwork and concrete filling. The formwork was made of two crossed U-shaped forms with a steel tube installed inside. Then, the fresh concrete (Cement: Zhengzhou Tianrui Cement Co., Ltd., Zhengzhou, China)was filled between the steel tube and the formwork.

FRP attachment and wrapping (Adhesive: Nanjing Hitech Composites Co., Ltd., Nanjing, China). The specimen surface was polished after curing for 28 days. First, the FRP cloth was vertically attached on the tension side. Second, the whole specimen was wrapped in 2 layers of FRP while making sure that the overlap area of FRP was as far as possible from the compression side. In addition, the two ends of each column were also wrapped by a CFRP (carbon-fiber-reinforced polymer) tie 100 mm in width for the purpose of premature failure of ends. 

Strain gauges attachment of FRP cloth. Four strain gauges 20 mm in length were attached on the mid-span surface of four sides of FRP cloth and used to test the axial strain of the FRP, as shown in Figure 2.

### 2.3. Test Procedure

Before the eccentric compression test, a column specimen with a steel cushion was placed on a knife-edge hinge, which was glued on the central testing platform by epoxy resin, as shown in Figure 3a, b. The column was capped with a steel plate. Then, a steel prism with a size of 160 mm × 35 mm× 25 mm was placed on the steel plate. In order to make sure that the eccentric position was accurately controlled by the designed distance, the center of the steel prism, the eccentric axis of the column and the center of the knife edge were kept in a line. The distance between the eccentric axis and the central axis of the column was the eccentricity. The designed eccentricities of all specimens are listed in Table 1.

During the test, three displacement meters (Hebei Xingtai Qiaoxi Kehua Resistance Strain Gauge Factory) were placed on the tension side along the axial direction of the column to measure the lateral deformation at different heights. A preliminary load of 10% ultimate load was carried out (Testing machine: Shanghai Shenke Testing Machine Co., Ltd., Shanghai, China) before the formal test for all specimens. The loading mode adopted was loading-until-damage in the formal test. The specimens were loaded at increasing load level with a 40 kN gap before cracking and with a 20 kN gap after cracking. The damage was defined as FRP rupture, concrete crush or a sharp decrease in load, whichever came first. The strains of the steel tube and FRP cloth, as well as the lateral deformation of the column, were recorded synchronously by the data acquisition system (Solartron Metrology, Leicester, UK). 

## 3. Test Results and Discussion

### 3.1. Failure Mode

In the process of the eccentric loading test, the FCSST columns were subjected to the combined action of axial force and axial moment. Accordingly, deformations, including axial compression and bending, were developed, as shown in Figure 4. With the increasing lateral deformation, the actual load eccentricity or bending moment changed along the axial direction and deviated from the initial value. When the maximum lateral deformation was reached, the bending moment was also the greatest. When the eccentricity was 5 mm, no tensile stress occurred in the initial stage of loading. However, with the growing load, tensile cracks appeared on the surface on the far side from the loading line. These cracks initially appeared as white spots on the FRP surface, suggesting that concrete wrapped by FRP starts to crack, which leads to damage of the external FRP resin matrix. With the development of tensile cracks, the lateral bending and deformation of specimens were sharply accelerated, and it further increased the actual load eccentricity far from the end section of the columns. Eventually, the specimens were damaged as a result of the expansion of the internal concrete on the compression side of column close to the loading line under compression and the consequent tensile fracture of the FRP cloth. It was observed from the failure mode of specimens that the greater the eccentricity, the greater the bending deformation.

Based on the test observations and relative studies [5], the failure mode was seen to be mainly affected by the eccentricity. The following description and Figure 4 detail the damaged areas of specimens.

When specimen A0E0 failed, one narrow strip (2 cm in width) of FRP cloth at around 1/5 of the height from the top loading end broke from the corner, while one wide strip (5 cm in width) of FRP cloth at around 1/4 of the height from the bottom loading end of the loading side also broke from the corner. According to the damage condition (Figure 4a), there might have been a little eccentricity during loading. 

The failure modes of specimens A0E5-a (Figure 4b), A0E5-b (Figure 4c), A1E5-a (Figure 4d), A1E5-b (Figure 4e), A1E15-a (Figure 4f), A1E15-b (Figure 4g), A2E15-a (Figure 4h), A2E15-b (Figure 4i), A1E30-a (Figure 4j), A1E30-b (Figure 4k), A2E30-a (Figure 4l), A2E30-b (Figure 4m), A2E45-b (Figure 4q) and A3E45-a (Figure 4r) were relatively positive. The failure area was at the middle–upper part (ranging from a 1/4 to a 1/2 of the height) of the compression side. As the concrete in the compression zone was crushed, the FRP at the corner was pulled apart from the compression side to its adjacent two sides. No significant damage to the FRP on the tension side was observed. However, the specimens of A1E45-a (Figure 4n), A1E45-b (Figure 4o), A2E45-a (Figure 4p) and A3E45-b (Figure 4s) exhibited damage to the concrete and FRP in the top areas (around 1/4 of the height from the top end) of the compressive zone.

### 3.2. Evolution of Axial Strain along the Cross-Section

The axial strains along the cross-section of the columns were measured through the strain gauges attached on the surface of the FRP cloth and steel tube (see Figure 5) at the intermedial height of the specimens. The evolution of axial strains at different positions (see Figure 5) and at different loading levels is shown in Figure 6. The compression strain was positive, and the tensile strain was negative in this study. In Figure 6, the first letter of the strain gauge’s name represents the material measured; F and S represents the FRP and steel tube, respectively.

Generally, the strains at different positions from the center line were almost linear at different loading levels, especially at the initial loading. With the growth of the loading, the neutral axis continuously moved towards the center of the cross-section, which led to the occurrence of tensile strain, that is, tensile cracks appeared on the side far from the loading line. Strain gauge F75-T of the FRP, which crossed the crack, recorded a great tensile strain. As shown by specimen A2E45-b in Figure 6h, the strain gauge F75-T was damaged by the development of a crack before reaching the maximum loading at 526.8 kN. Strain gauge F75-C of the FRP generally showed a small value, such as that for specimen A1E15-a, shown in Figure 6. It was observed that the circular tensile failure of the FRP on the compression side occurred at a position far from the strain gauges. Moreover, when the strain value of the steel tube exceeded 0.002, they were obviously no longer in linear (see Figure 6a–h), which may have resulted from the non-uniform local plastic deformation of the steel tubes.

Through the comparison of F75-C and F75-T strains of the FRP on the compression side and tension side, it was found that, when the eccentricity was smaller than 15 mm, the increase in F75-C and F75-T strain values was close to linear at the initial loading (see Figure 6a–c). This suggests that the specimens experienced an elastic deformation, and their neutral axes of the cross-section did not change yet. When the strain value reached 0.002, the compressive strain recorded by F75-T dropped sharply and then changed into tensile strain. It shows that, with the growth of the loading, the neutral axis gradually moved to the center of the cross-section. As the eccentricity increased, the initial transformation of the F75-T strain was reduced gradually (see Figure 6d). When the eccentricity was greater than 30 mm, the F75-T completely showed a tensile strain, as shown in Figure 6e–h.

### 3.3. Axial Bearing Capacity

The evolution of the ultimate bearing capacity of FCSST columns with varying eccentricity is shown in Figure 7. Due to the existence of a large bending moment, the axial compression bearing capacity of the FCSST columns with large eccentricity was small. This result is consistent with the findings relating to circular FRP–concrete–steel composite columns [5]. With the growth of the eccentricity, the axial bearing capacity dropped sharply from 991.9 kN to 430 kN, which shows a decrease rate of 50%. Additionally, as the FRP cloth on the tension side increased from one layer (e.g., A1E15-a, A1E15-b, A1E30-a, A1E30-b, A1E45-a and A1E45-b) to two layers (e.g., A2E15-a, A2E15-b, A2E30-a, A2E30-b, A2E45-a and A2E45-b), the corresponding ultimate bearing capacity increased slightly from 3.6% to 13% on average. Therefore, compared to the columns without FRP cloth [5], a column reinforced by FRP cloth on the tension side and subjected to eccentric load can effectively increase its load capacity.

Compared with the effect of various FRP cloth layers, eccentricity had a more remarkable impact on the compression bearing capacity of the FCSST columns. In the case of low eccentricity, the failure zone of specimens (e.g., A0E0, A1E5-a and A1E5-b) with low bearing capacity deviated from a certain range around the intermedial height of the columns. The axial bearing capacity of specimen A0E0 was 991.9 kN, while it reached 1032.4 kN for specimen A1E5-a with a one-layer FRP cloth, although it was subjected to a loading with 5 mm eccentricity. Meanwhile, obvious local concrete crushing was observed. Under axial eccentric load and with an increase in lateral deflection, the actual eccentricity will vary along the height of specimens, which deviates from the initial value with the growth of the loading. In addition, the local deformation near the failure zone will enhance the growth of the eccentricity.

In this study, it was the case that two specimens of the same group (which were supposed to be the same) had a significantly different bearing capacity. Comparing the four specimens with *e* = 15 mm, the bearing capacity of A1E15-a and A2E15-a was obviously greater than that of A1E15-b and A2E15-b. The reason was that the damaged area of A1E15-b and A2E15-b was relatively far away from the intermedial height of the columns. Significant local deformation occurred, with the concrete being crushed and the FRP being pulled apart. Under the same load, the actual bending moment changes along the height of specimens, and the changing may be different even for the same specimen with the same eccentricity. In addition, under the same load, the specimen subjected to a greater actual eccentricity will have a relatively larger bending moment, which makes the bearing capacity decrease in different degrees correspondingly. That was also true for the six specimens (e.g., A1E45-a, A1E45-b, A2E45-a, A2E45-b, A3E45-a and A3E45-b) with *e* = 45 mm. The only difference for them was that the damaged area was further away from the cross-section at the intermedial height of columns but was closer to the reinforced zone at the end. Some specimens (e.g., A1E45-a) even only had local damage in the reinforced area.

According to Table 1 and the above analysis, it can be seen that the damaged area has a significant influence on the bearing capacity of specimens. Therefore, measures should be taken to make sure that damages are controlled in a certain range of the cross-section at the intermedial height of the columns in practice. Meanwhile, sufficient FRP cloths should be used to strengthen the ends to avoid the local crushing of column ends in the compression area.

### 3.4. Axial Load–Lateral Mid-Span Deflection Curve

The axial load–lateral mid-span deflection curves of the FCSST columns are presented in Figure 8. It can be seen from Figure 8a that, in the case of eccentricity *e* = 5 mm, the column (e.g., A1E5-a) wrapped by a one-layer FRP cloth on the tension side produced the curve that reached the peak phase after an approximate linear elastic stage and then went through a gentle decrease stage, which suggests that the column generally lost its bearing capacity. It can be seen from Figure 8a,b that, compared with the curve of the columns with 5 mm eccentricity, when the *e* increased up to 15 mm and above, the curves of those columns (e.g., A1E15-b, A1E30-b, A1E45-b, A2E15-a, A2E30-b and A2E45-b) wrapped by one or two layers of FRP cloth on the tension side showed a completely different trend. That is, the whole curve presented a parabola that first increased gradually and then decreased slowly. The curves of those with *e =* 15 mm and above showed a more moderate slope at the initial ascent stage. The slope declined gradually with the growth of *e* eccentricity compared with that of the column with 5 mm eccentricity and changed continually until it entered a flat strengthening stage. Except for the curve of the column with 5 mm eccentricity, all curves of the columns with different eccentricity were basically similar. The only difference was that, under the same load, the deflection with the growing *e* increased gradually, while the area under the curve decreased, suggesting a declining energy dissipation capacity of the columns [41].

With the different layers of FRP cloth and at the same eccentricity, it can be seen from Figure 8c–f that the columns produced the curves with a similar trend. As the FRP cloth layer increased, the curves showed more rigid characteristics, and the load increased to different degrees under the same deflection. As the loading continued, the FRP cloth on the tension side was more effective in preventing the development of lateral deflection under the same load. It further shows that the FRP cloth attached on the tension side can improve the ultimate bearing capacity to some extent.

It can be seen that the FCSST columns obtained a larger axial load than the traditional steel-bar-reinforced concrete (steel–RC) column specimens [34]. In addition, although the FCSST columns had a hollow core for the reason of low self-weight, they exhibited a lower average reduction in ductility under eccentric axial loads than the steel–RC specimens [34].

### 3.5. Evolution of Lateral Deflection Curve along the Height of Columns

In order to present the lateral deflection development more visually, the evolution curves of lateral deflection along the height of the columns under different eccentricity were drawn, as shown in Figure 9, where *H* is the height from the measuring point to the bottom of columns with the growth of *e* eccentricity, and *n* is the ratio of the applied load to peak load. When the eccentricity was 5 mm, it was found that, as in Figure 9a,b, at the initial loading stage, the compression strain first occurred at the places far from the loading side. With the growth of the load, the compression strain decreased gradually and then the tensile strain finally appeared. For the column with one layer of axial FRP cloth, its compression deformation decreased significantly at the initial loading (e.g., A1E5-a in Figure 9c). When *e* = 15 mm, 30 mm and 45 mm, the evolution curves of lateral deflection along the height of columns were similar to those of previous eccentricity studies [5,41]. That is, the greater the load, the greater the deflection. In addition, FRP cloth on the tension side can greatly reduce the deflection of specimens under the same load and improve the resistance to deformation of columns.

Most of the lateral deflection curves in this study showed a difference from the results of previous eccentricity studies [1,42,43]. That is, the upper and lower deflection were asymmetric, and the upper deflection was the greatest. As the ball hinge of the machine that attaches the top of square columns can move freely, slippage could more easily happen. However, the bottom of the columns was connected to the knife-edge hinge, thus it was relatively difficult for slippage to occur. Moreover, compared with medium and long columns, the height of the columns in this study was shorter, and the difference between the upper and lower hinge support was not easy to reduce [44], leading to the asymmetry of the upper and lower deflection.

## 4. Conclusions

The experimental results and failure mode of 19 FRP–concrete–steel double-skin tubular (FCSST) square columns subjected to eccentric loading were investigated. The failure mechanism, the evolution of axial loading along the cross-section and the change in lateral deflection along the height of the columns were discussed. In addition, the influence of eccentricity and axial FRP cloth on the bearing capacity and axial load–lateral deflection curves of the columns was analyzed. The following conclusions were drawn:(1)Under eccentric loading, the axial strain of the cross-section at the intermedial height of columns presents a nearly linear development along different distances from the center line, especially in the initial loading stage;(2)The axial compression bearing capacity of columns with large eccentricity is small. With the increase in axial FRP cloth on the tension side, the corresponding ultimate bearing capacity tends to increase gradually;(3)When eccentricity is small (*e* = 5 mm), the axial load–lateral deflection curve of columns is similar to that of axial compression. When eccentricity is large (*e* = 15, 30 and 45 mm), the development of the axial load–lateral deflection curve is similar. Additionally, their area under the curve decreases gradually, which indicates that the ability to absorb energy declines correspondingly. Therefore, the axial FRP cloth on the tension side can well prevent growing lateral deflection;(4)Based on the test results and analysis, two suggestions for the structural design and testing method are provided as follows: (1) two-way FRP cloth can be used for the reinforcement of columns, especially for the reinforcement of the tension side of columns under eccentric loading; (2) in order to fully utilize the performance of FCSST columns and avoid local premature failure, the two ends (top and bottom) of the column should be reinforced by FRP cloth before testing.

## Figures and Tables

**Figure 1 polymers-15-02642-f001:**
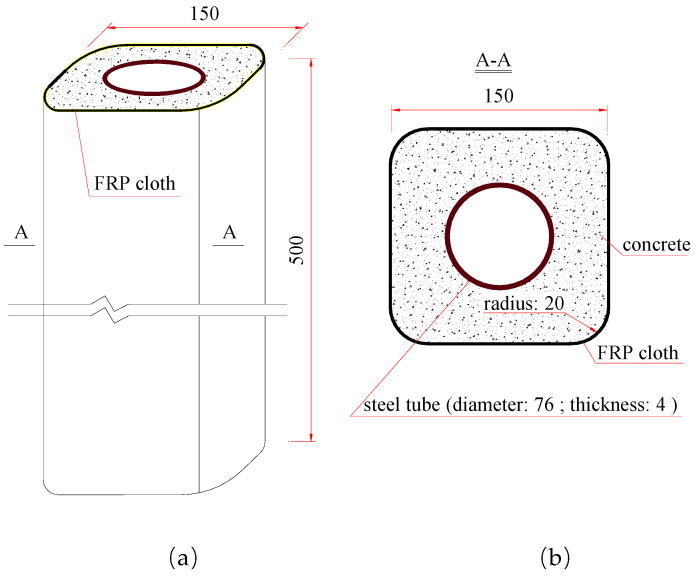
Schematic diagram of (**a**) FCSST column (mm) and (**b**) its cross-section (mm).

**Figure 2 polymers-15-02642-f002:**
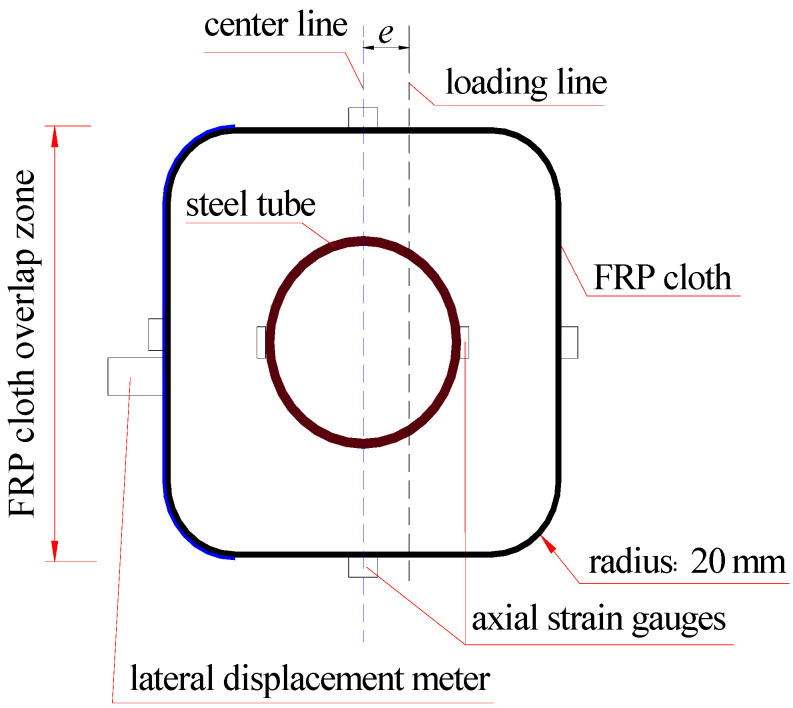
Schematic layout on the mid-span of FCSST column.

**Figure 3 polymers-15-02642-f003:**
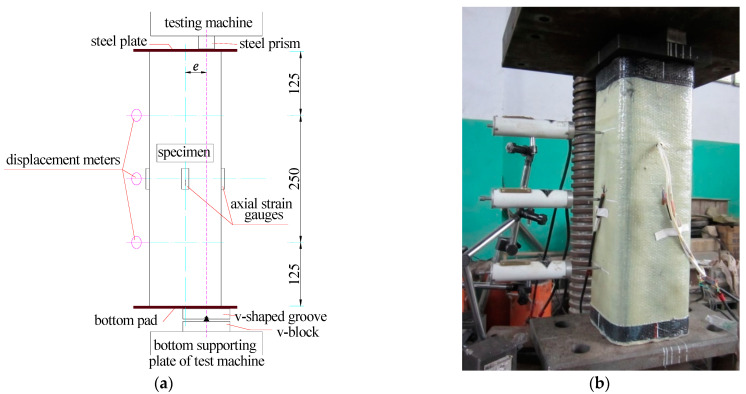
(**a**) The loading scheme (mm) and (**b**) the testing site of eccentric compression test for FCSST columns.

**Figure 4 polymers-15-02642-f004:**
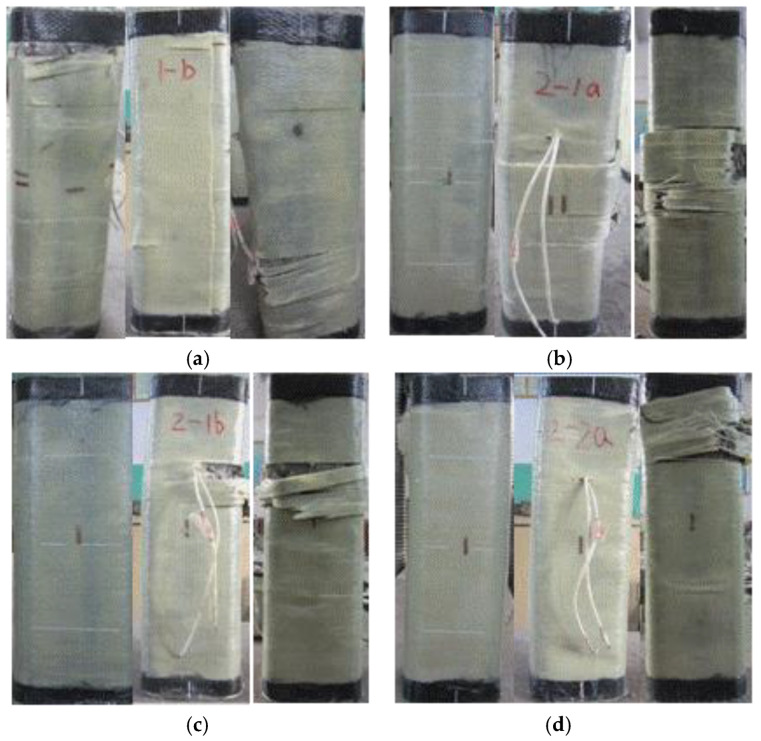
Photos of tested FCSST columns: (**a**) A0E0, (**b**) A0E5-a, (**c**) A0E5-b, (**d**) A1E5-a, (**e**) A1E5-b, (**f**) A1E15-a, (**g**) A1E15-b, (**h**) A2E15-a, (**i**) A2E15-b, (**j**) A1E30-a, (**k**) A1E30-b, (**l**) A2E30-a, (**m**) A2E30-b, (**n**) A1E45-a, (**o**) A1E45-b, (**p**) A2E45-a, (**q**) A2E45-b, (**r**) A3E45-a and (**s**) A3E45-b.

**Figure 5 polymers-15-02642-f005:**
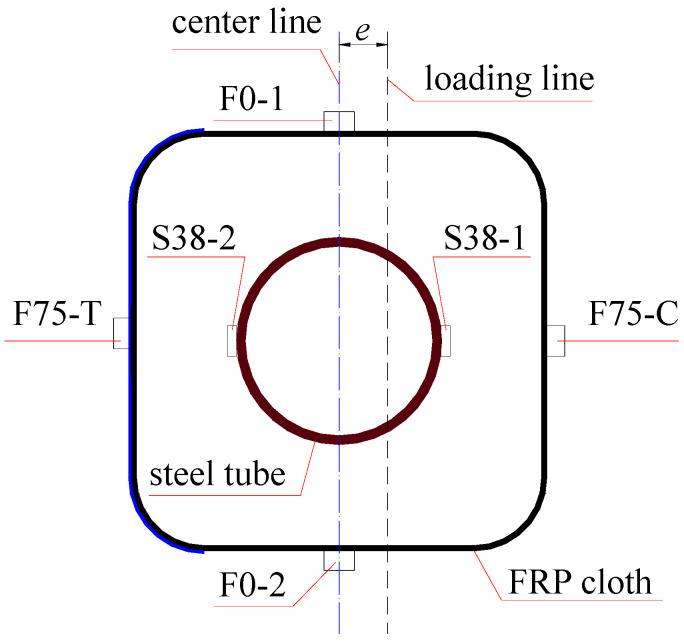
Axial strain gauge location along the cross-section of FCSST columns. Note: for the strain gauge names, the first letters of F and S represent the gauge attached on the FRP and steel tube, respectively; the following number is the distance from the center line to the gauge position; the last letters T and C represent the tension and compression, respectively, e.g., F75-T represents the strain gauge attached to the FRP and located in the tensile zone with a distance of 75 mm from the center line. F0-1 and F0-2 represent the two strain gauges attached on the FRP and located on the center line.

**Figure 6 polymers-15-02642-f006:**
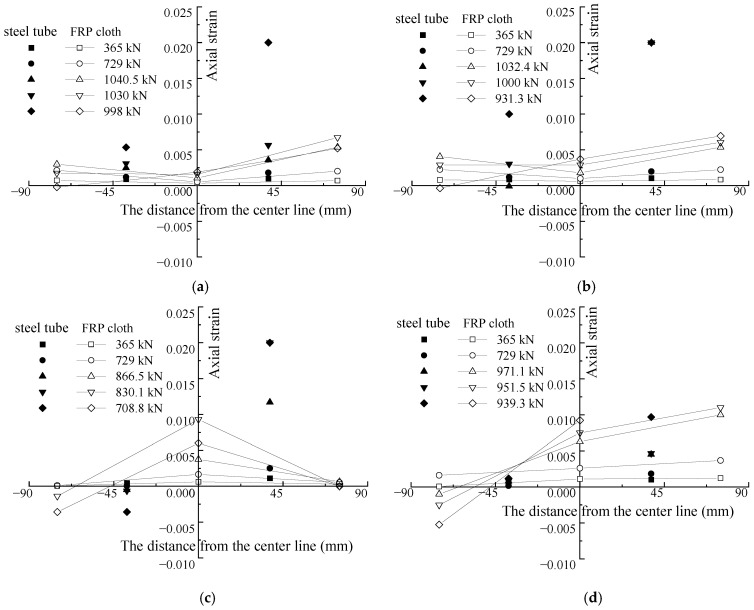
Axial strain evolution along the cross-section of FCSST columns: (**a**) A0E5-b, (**b**) A1E5-a, (**c**) A1E15-a, (**d**) A2E15-a, (**e**) A1E30-b, (**f**) A2E30-b, (**g**) A1E45-a and (**h**) A2E45-b.

**Figure 7 polymers-15-02642-f007:**
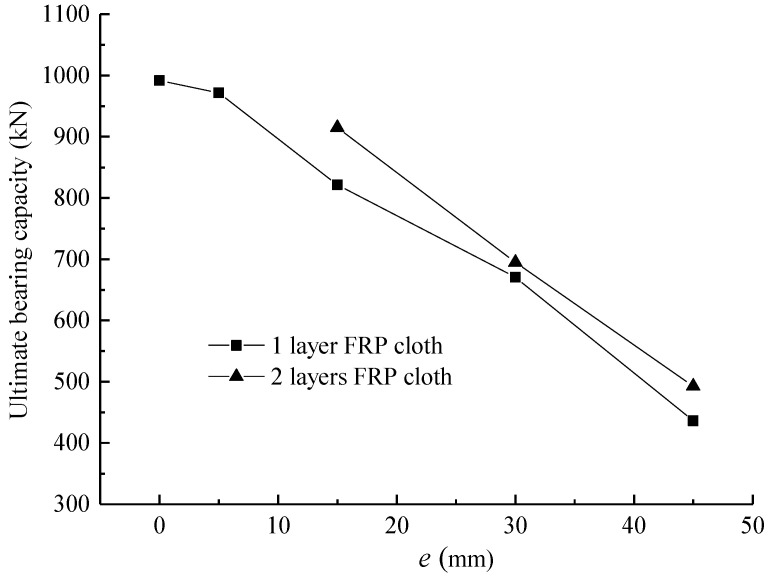
Evolution of ultimate bearing capacity of FCSST columns with varying eccentricity.

**Figure 8 polymers-15-02642-f008:**
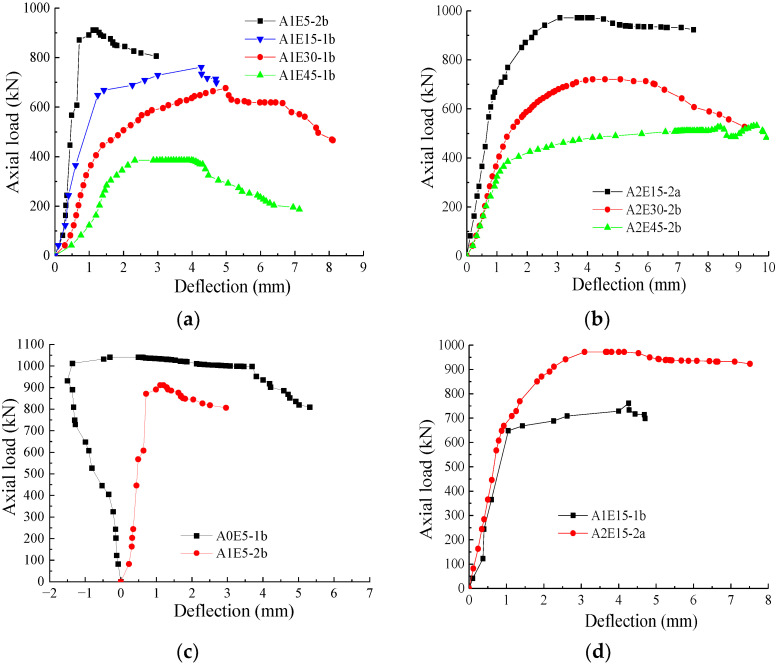
Axial load–lateral deflection curves of FCSST columns: (**a**) different eccentricity (1-layer FRP cloth), (**b**) different eccentricity (2-layer FRP cloth), (**c**) different layers of FRP cloth (*e* = 5 mm), (**d**) different layers of FRP cloth (*e* = 15 mm), (**e**) different layers of FRP cloth (*e* = 30 mm), (**f**) different layers of FRP cloth (*e* = 45 mm).

**Figure 9 polymers-15-02642-f009:**
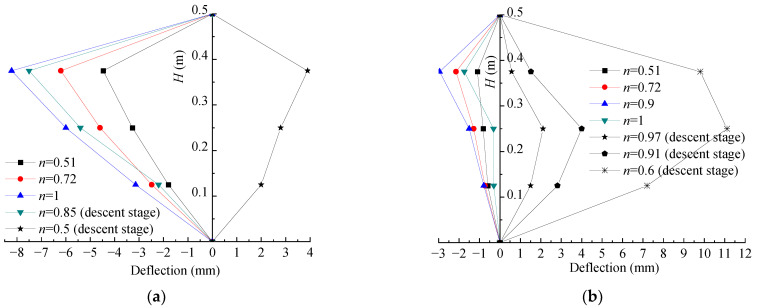
Evolution of deflection along the height of FCSST columns: (**a**) A0E5-a, (**b**) A0E5-b, (**c**) A1E5-a, (**d**) A1E15-b, (**e**) A1E30-b, (**f**) A2E30-b, (**g**) A1E45-a and (**h**) A2E45-b.

**Table 1 polymers-15-02642-t001:** Specimen design and test results.

Name ^1^	Layers of Axial FRP ^2^	Eccentricity (mm)	Strength of Concrete Cube (MPa)	Peak Load (kN)	Failure Mode ^3^
A0E0	0	0	54.9	991.9	①
A0E5-a	0	5	53.6	1068.8	②
A0E5-b	0		53.9	1040.5	②
A1E5-a	1		55.8	1032.4	②
A1E5-b	1		52.7	911.0	②
A1E15-a	1	15	50.6	886.8	②
A1E15-b	1		56.4	761.4	②
A2E15-a	2		54.4	971.7	②
A2E15-b	2		54.9	858.5	②
A1E30-a	1	30	55.3	664.3	②
A1E30-b	1		53.0	676.5	②
A2E30-a	2		52.6	668.4	②
A2E30-b	2		57.1	720.9	②
A1E45-a	1	45	52.1	486.4	③
A1E45-b	1		49.9	385.3	③
A2E45-a	2		48.3	450.0	③
A2E45-b	2		50.8	534.9	②
A3E45-a	3		55.0	474.2	②
A3E45-b	3		56.9	425.7	③

^1^: The specimens are named in the format of A*i*E*j*, where A*i* represents the axial FRP cloth with *i* layers, and E*j* represents the eccentricity, i.e., A2E15 represents the specimen with an axial FRP cloth with 2 layers tested in an eccentricity of 15 mm. The specimens of each series numbered a and b were exactly the same. ^2^: The axial FRP cloth was only arranged on the tension side. All specimens were wrapped in 2 layers of hoop FRP after axial FRP cloth attachment; the control specimen of A0E0 was directly wrapped in 2 layers of hoop FRP. ^3^: Failure mode: ① lower part of the column and not the overlap area, FRP tensile failure at corner; ② the middle–upper part on the compression side of column, FRP tensile failure at corner or ribboned; ③ the upper part on the compression side of column, FRP tensile failure at corner and crashed concrete at the end part.

## Data Availability

Data are contained within the article.

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
