# Peer review of "Eccentric Compression Properties of FRP–Concrete–Steel Double-Skin Square Tubular Columns"

_polymers, 2023, doi:10.3390/polym15122642_

Round 1

Reviewer 1 Report

In this study, authors prepared FRP-concrete-steel double skin square tubular columns and determined its compression properties. I urge the authors to address the following comments.

1. Please do not use abbreviation in title of manuscript.

2. Please write the expanded form of abbreviation where use firstly in manuscript like in firs line of abstract.

3. What do you mean remove free template?

4. What is the new and novel in this study beyond the already reported literature?

5. Which polymer was used in FRP cloth and what was the thickness of that cloth?

6. Please support the results with literature and show how your material is better than previously prepared composites?

7. Please cite some latest papers of the respective journal.

8. There are some grammatical mistakes in the manuscript. Please revise it carefully.

There are some grammatical mistakes in the manuscript. Please revise it carefully.

Author Response

  1. Please do not use abbreviation in title of manuscript.

Response: The abbreviation FRP has displayed its full name “Fiber Reinforced Polymer” in the following brackets.  Please see the yellow-highlighted words of the title.

  1. Please write the expanded form of abbreviation where use firstly in manuscript like in firs line of abstract.

Response: The abbreviation FRP has displayed its full name “Fiber Reinforced Polymer” in the following brackets. Please see the yellow-highlighted words of abstract. Similar problems have been checked for the rest of this manuscript.

3.What do you mean remove free template?

Response: Sorry for the wrong expression. It should be “permanent formwork” and has been revised, please see the yellow-highlighted words of abstract.

  1. What is the new and novel in this study beyond the already reported literature?

Response:

Compared with those have been published, the novel of this study lies in square columns, rather than the circular columns which has been widely studied.  Moreover, the FRP-concrete-steel column has great potential in application, but the related studies are still limited. Those have been reorganized and shown in the last two paragraphs of Introduction.

Besides the novel mentioned above, this study also has another three novel aspects: 1. The composite of FRP, concrete and steel is adopted in columns, which takes full advantage of those positive performances in columns; 2. For the propose of low cost, the FRP cloth is only used on the tensile side, and its layer increases with the increasing eccentricity; 3. The eccentricity designed in this study covers a larger range of relative eccentricity. 

The three aspects have been supplied in Introduction and highlighted in yellow.

  1. Which polymer was used in FRP cloth and what was the thickness of that cloth?

Response: The polymer used in FRP cloth is Glass Fiber Reinforced Polymer, please see the highlighted words of the first line in section 2.1.

The thickness of that cloth is 0.17 mm.  The supplement was added in the first paragraph of section 2.1, please see the yellow highlighted words.

  1. Please support the results with literature and show how your material is better than previously prepared composites?

Response: Compared to the columns without FRP cloth [5], the column reinforced by FRP cloth on tension side and subjected to eccentric load can effectively increase its load capacity.

It can be seen that the FCSST columns obtained a larger axial load than the traditional steel bar reinforced concrete(steel-RC) column specimens [34]. In addition, although the FCSST columns have a hollow core for the reason of low self-weight, they exhibited a lower average reduction in ductility under eccentric axial loads than the steel-RC specimens [34].

Above supplements have been inserted in paragraph 1 of section 3.3 and paragraph 3 of section 3.4, please see the yellow highlighted sentences.

  1. Please cite some latest papers of the respective journal.

Response:  Reference [32-35] have been added and highlighted in yellow.

  1. There are some grammatical mistakes in the manuscript. Please revise it carefully.

Response:  All mistakes has been checked and revised.

Reviewer 2 Report

The paper presents a study on eccentric behaviour of FRP-concrete-steel doubleskin square tubular columns, on the base of experimental tests. The paper is well-structured and well-written and, in my opinion, should be accepted for publication. Few comments are following provided, in order to improve some aspects:

- In the introduction, authors present details about this particular kind of column, which anyway is not common within the existing building stock. I suggest to better frame the problem steatement, providing some insights about when this type of column is used. Probably, being the main problem the eccentricity, the seismic problem could be employed and for this scope, general studies could be mentioned, as 10.1007/s10518-022-01516-7 and references therein.

- In section 2, I suggest to add some information about the FRP selection, used for the specimens. 

- Figure 6 should be re-formatted

- Photos about failures for the columns could be added

- Did authors perform other tests to compare the obtained behaviour with a basic configuration of this column typology (e.g., without FRP)?

- In the end, what about the possible implications in a view of Technical code proposal? What can authors suggest? 

Author Response

  1. In the introduction, authors present details about this particular kind of column, which anyway is not common within the existing building stock. I suggest to better frame the problem steatement providing some insights about when this type of column is used. Probably. being the main problem the eccentricity, the seismic problem could be employed and for this scope, general studies could be mentioned as 10.1007/s10518-022-01516-7 and references therein.

Response: The composite of FRP, concrete and steel adopted in columns can take full advantage of those positive performances in columns. Accordingly, this composite of FRP-concrete-steel in column has great mechanical behavior, especially for columns under eccentric compression. Therefore, this column still has huge potential in application, i.e. pier in corrosive environment, support in industrial and civil buildings, etc. [15,21,24,25]. Besides, its good seismic resistance could guarantee its application in building structures with seismic requirement [26-30].

Above supplement has been inserted in Paragraph 5 of Introduction and highlighted in yellow or green. Moreover, the references you provided is helpful for my study and added in Ref. [26-30]

  1. In section 2, I suggest to add some information about the FRP selection, used for the specimens.

Response: Since the glass fiber reinforced polymer (GFRP) has advantages of low cost, easy fabrication and high tensile performance, it was used for the FRP cloth in this study.

Above supplement has been inserted into the first sentence of Section 2.1 and highlighted in green.

  1. Figure 6 should be re-formatted.

Response: Fig. 6 has been re-formatted.

  1. Photos about failures for the columns could be added.

Response: More photos about failures for the columns have been in Fig.4 and highlighted in green.

  1. Did authors perform other tests to compare the obtained behaviour with a basic configuration of this column typology (e.g., without FRP)?

Response: A0E0 is a control specimen without FRP for 0 mm eccentricity. Its results have been listed and green highlighted in Table 1. Its analysis with others has been green highlighted in paragraph 3 of Section3.1, paragraph 2 of Section3.3 and Fig. 4.

Unfortunately, the control specimens without FRP for 15/30/45 mm eccentricity were not investigated and will be studied in the near future.

  1. In the end. what about the possible implications in a view of Technical code proposal? What can authors suggest?

Response: Based on the test results and analysis, two suggestions for the structural design and testing method are provided as follows: 1) the two-way FRP cloth can be used for reinforcement of column, especially for the reinforcement of tension side of column under eccentric loading; 2) In order to fully utilize the performance of FCSST column and avoid the local premature failure, the two ends (top and bottom) of the column should be reinforced by FRP cloth before testing.

The supplement is placed and green highlighted into Conclusions, please see the last paragraph of Conclusions.

Round 2

Reviewer 1 Report

Comments have been addressed adequately.